# New Liquid Source of Antioxidant Phenolic Compounds in the Olive Oil Industry: Alperujo Water

**DOI:** 10.3390/foods9070962

**Published:** 2020-07-21

**Authors:** María África Fernández-Prior, Juan Carlos Pérez Fatuarte, Alejandra Bermúdez Oria, Isabel Viera-Alcaide, Juan Fernández-Bolaños, Guillermo Rodríguez-Gutiérrez

**Affiliations:** Instituto de la Grasa, Consejo Superior de Investigaciones Científicas (CSIC), Campus Universitario Pablo de Olavide, Edificio 46, Ctra. de Utrera, km 1, 41013 Seville, Spain; mafprior@ig.csic.es (M.Á.F.-P.); juancarlos.perez@monteec.com (J.C.P.F.); aleberori@ig.csic.es (A.B.O.); iviera@ig.csic.es (I.V.-A.); jfbg@cica.es (J.F.-B.)

**Keywords:** olive oil, phenolic extracts, alperujo, hydroxytyrosol, 3,4-dihydroxyphenylglycol, antioxidant, functional foods

## Abstract

The current management of alperujo as the main solid by-product from the two-phase olive oil extraction system has led to the appearance of a new liquid effluent that until now was treated together with the alperujo itself. The composition and antioxidant properties of its bioactive components at different depths of the pond were studied using colorimetric and HPLC with UV and MS detectors, DPPH, reducing power and rancimat. The concentration of suspended solids varied between 1.71 and 8.49 g/L, total fat was between 0.74 and 1.47 g/L, and total phenols were found between 3.74 and 4.11 g/L, which included hydroxytyrosol, 3,4-dihydroxyphenylglycol and tyrosol as the main phenols. Two types of extracts were obtained through two industrial systems with ethyl acetate and by chromotography, with an average content in total sugars of 2.1% and 3.16%, total phenols of 17.9% and 28.6% and hydroxytyrosol of 51.5 and 79.0 mg/g of extract, respectively. The activity presented by the chromatographic extract was higher in terms of free radical sequestering capacity, reducing power and the inhibition of lipid oxidation. Obtaining bioactive extracts would improve the formulation of food with natural components and at the same time would be the first step in a biorefinery to improve the management of the new effluent.

## 1. Introduction

The two most widely used continuous extraction systems in the world for the extraction of olive oil are the three-phase extraction system, in which two by-products are generated, one liquid (alpechín) and one solid (orujo), and the two-phase extraction system in which only one semi-solid by-product called alperujo is generated. The olive oil industry in Spain generates alperujo as its main solid by-product from the two-phase olive oil extraction system. Spain is the greatest producer with a range of 1.2–1.5 MTn/year of olive oil in the last three years [1]. The amount of alperujo generated per year is about 4.8–6 MTn. Alperujo is a semisolid substance with a variable percentage of humidity (60–80%) and high organic content, which makes it very difficult to manage. Initially, when the three-phase olive oil extraction system was replaced by the two-phase system in Spain in the nineties, the alperujo, with a greater content in humidity, replaced the solid waste called orujo (50–55% humidity), thus increasing the environmental and technical problems concerning alperujo management [2]. Nowadays, the main use of alperujo is energy production after the extraction of pomace olive oil [3]. Both for the extraction of the pomace oil and for its final use as fuel, the alperujo must be dried below 10% humidity, which considerably raises the industrial costs [4]. Recycling this kind of compounds from the olive oil industry is in line with the global trends of environmental protection and transformation of waste into useful products that can be used in agriculture, chemical, textile, pharmaceutical and food industries [5]. Many alternatives are emerging at the research level to achieve the reuse of all olive oil by-products. Most of them entail the extraction and recovery of the bioactive components, phenols [6]. Because the major part of the phenols present in the olive fruit (about 98–99%) remain in the alperujo after olive oil extraction, the extraction of bioactive compounds from it is gaining importance as an industrial alternative [7]. Two of the most remarkable water-soluble phenols present in alperujo are hydroxytyrosol (HT) and the 3,4-dihydroxyphenylglycol (DHPG) because of their biological activities [8,9]. HT is one of the most widely studied phenols in the olive for its biological potential, which goes beyond its powerful antioxidant activity, and presents important properties in the prevention against coronary diseases, diabetes, certain types of cancer and inflammatory diseases, among others [10]. Many studies have been focused on the extraction of HT by different methods or even by chemical synthesis [11]. In the case of DHPG, it is a phenol which is very similar in structure to HT and with similar properties. In fact, it has recently been determined that it even exceeds some of the HT activities and in many cases has been found to have a powerful synergistic effect with HT [12,13]. Until a few years ago, this phenol was known to be used as a biomarker of neurodegenerative diseases and a metabolite of noradrenaline [14]. Thanks to its recent development, there are sufficient quantities available to determine its biological potential, such as the prevention of colitis or its anti-inflammatory effect [15]. One of the important properties recently attributed to DHPG is its ability to act as a phytoregulator in the germination and growth of industrial crops in agriculture [16]. The growing demand both at the research and commercial level makes it necessary to search for new liquid sources which facilitate the extraction of these phenols at a reduced cost. The problem is that to extract the phenols from alperujo, it is necessary to apply thermal, chemical or enzymatic treatments that make the process more expensive or even degrade some of the bioactive compounds present such as DHPG. Obtaining new sources would contribute to a better development of HT and DHPG within the field of food and agriculture.

According to the traditional method, alperujo is collected in the pomace olive oil extractors and stored in huge open air ponds before it is dried and submitted for the extraction of the residual oil. During storage in the open air ponds, the concentration in phenolics does not decrease as a consequence of the subsequent drying [17]. Because of the high humidity enhanced by the addition of liquid effluents in olive mills, such as vertical centrifuge waters, water ponds are formed on the surface of alperujo during storage [3]. Until now this water, or alperujo water, was homogenized with the alperujo to be treated in the dryer. Many pomace extractors are now starting to remove this water to facilitate the pumping of the alperujo and its subsequent drying, and it is stored in other open air ponds before its evaporation and depuration. The appearance of this liquid effluent improves the drying of the alperujo but implies a greater environmental impact due to the need to build new evaporation ponds with their own environmental problems. In this water, part of the water-soluble compounds present in the alperujo are solubilized, such as phenolic compounds, which are a potential source of bioactive compounds that could easily be extracted without any pre-treatment. This water is not similar to the water from the three-phase olive oil extraction system (alpechín) because, having been in contact with the pulp for much longer, it has a much higher organic load, and represents a new source of more interesting bioactive compounds than alpechín. On the other hand, phenolic compounds are phytotoxic components that make it difficult to use biological systems for their purification. Therefore, the recent appearance of this effluent may imply serious environmental and management drawbacks at the same time that it could be an interesting source of bioactive compounds. The aim of this work is to study the composition and the extraction of the added-value compounds as a first step of the biorefinery of alperujo water. The extraction of the phenolic compounds diminishes the toxicity of this water, making it possible to apply a further bioprocess to obtain the total utilization or depuration of these new effluents [18].

The alperujo water is stored in 1.5 m deep ponds for up to four months during the olive oil extraction season. It is important to know how its phenolic composition, fat and amount of suspended solids are affected by an average storage period of two months in order to make industrial use of phenols. These factors are important for applying the technologies that currently operate in industry for the extraction of phenols, such as the use of organic solvents like ethyl acetate and chromatographic systems. The depth of the pond can cause a loss in phenolic compounds due to the concentration of oxygen or the incidence of UV light, which also influences the sedimentation of suspended solids and fat [17].

In the present work, we have studied a liquid source from alperujo ponds for the first time, where these phenols are present in soluble form and therefore are more accessible for extraction. We have also studied the effect of the depth of the pond on the phenolic, fat and suspended solid compositions of these alperujo waters in open air evaporation ponds. In addition, the application of two of the most commonly used systems has been studied in order to verify the industrial extraction from this new effluent, and to monitor the antioxidant potential in vitro of the extracts obtained.

## 2. Materials and Methods

### 2.1. Materials

The alperujo water samples were taken from the company Oleicola El Tejar Nuestra Señora De Araceli S.C. which is the industry that processes the largest amount of alperujo in the world through various industrial plants. The samples were taken from three of its main plants located in three different areas of Andalusia, which receive approximately half of the alperujo generated in this region, where more than 35% of the world’s olive oil is produced. Samples (25 L) were taken from three different depths in the ponds, the upper part, the middle part (0.75 m) and the lower part (1.5 m, (total depth)) from each of the three industrial plants. The three samples from each of the three depths were pooled together to be sufficiently representative. All of them were taken at the end of the 2018/2019 campaign to include alperujo from both the beginning and the end of the season. The storage time of the alperujo water varied, since it had been produced throughout the four months of the campaign, so the average storage time was two months. The samples were all transported on the same day they were taken and stored in 1 and 5 L containers in freezers at −20 °C until use.

### 2.2. Standard Compounds

3,4-dihydroxyphenylglycol was obtained from Sigma-Aldrich (Deisenhofer, Germany). Tyrosol was obtained from Fluka (Buchs, Switzerland) and hydroxytyrosol was obtained from Extrasynthese (Lyon Nord, Geney, France). Standards of gallic acid (GA), trifluoroacetic acid (TFA), anthrone, glucose, Folin-Ciocalteu’s phenol reagent, and 2,2-diphenyl-1-picrylhydrazyl were purchased from Sigma-Aldrich (Madrid, Spain). Sodium bicarbonate (Na_2_CO_3_), acetonitrile (HPLC grade, hexane and sulphuric acid (H_2_SO_4_) were obtained from PanreacQuimica S.A. (Barcelona, Spain).

### 2.3. Determination of Insoluble Solids

Approximately one liter of each sample, measuring exactly each volume, was centrifuged in six centrifuge cups at 15,000× *g* at 4 °C for 20 min in a Sorvall RC-5C refrigerated centrifuge (Du Pont Instruments, Newton, CT) and the solid was separated from the liquid. The solids were joined in a tared rack and left in the oven at 50 ºC until constant weight. Once dry, the grams-per-liter obtained from the suspended solids were quantified for the three samples collected at different depths of the pond.

### 2.4. Determination of Fat in the Separated Solid and Liquid

The solids obtained from the three samples were dried at 50 °C and crushed in a mortar to break them up. The samples were then weighed, and placed in a Soxhlet to extract the fat [17]. The difference between the initial and the de-fatted sample was determined by gravimetric analysis to calculate the fat content in each sample.

As for the fat content in the liquid, three successive extractions were made with cold hexane in a decanting funnel from 300 mL of the centrifuged liquid using the same volume of solvent as liquid in each extraction. After separating the aqueous from the organic phase, the hexane from each extraction was combined and introduced into the vacuum evaporator to eliminate the solvent. Finally, the fat content of the liquid was quantified gravimetrically.

### 2.5. Determination of pH and the Amount of Dry Matter Content of the Samples

The pH of each alperujo water sample was analyzed using a pH-meter model Crison 20 Basic [18]. The amount of dry matter content of the samples was gravimetrically determined. Between 0.5 mL and 2 mL of extract was air-dried and introduced in a desiccator until a constant weight was reached, with the result expressed as milligrams of extract per gram of fresh sample.

### 2.6. Phenolic Extracts

Two phenolic extracts were obtained at laboratory scale from the liquid fractions by two different methods commonly used at the industrial level:

Phenolic extract with ethyl acetate. An organic solvent extraction was carried out by using ethyl acetate in a heating counter-current extraction system for eight hours (Alamo-Ref.:00362000, Spain). For this purpose, 500 mL of ethyl acetate were used for every 200 mL of sample, totaling up to two liters of sample. Basically, the solvent, upon evaporation, passes from the bottom of the sample to the top, back to the heat source where it is again evaporated. Finally, all the ethyl acetate was evaporated under vacuum to obtain the phenolic extracts from each sample.

Phenolic extract obtained by liquid chromatography. An industrial patent was used to produce the second phenolic extract [19]. The liquid phase (2 L) was extracted by ionic resin previously washed with deionized water and activated with a 1% solution of orthophosphoric acid. Finally, the extract was eluted with water and concentrated in a vacuum evaporator at 60 °C, obtaining 5 g of final extract.

### 2.7. Determination of Single Phenols by HPLC

HPLC-DAD: The samples were analyzed by HPLC-UV and the main individual phenolics were determined, such as 3,4-dihydroxyphenol glycol, hydroxytyrosol and tyrosol (Ty), among others. All the aliquots of the aqueous extracts were filtered through a 0.45 µm membrane and injected. The analysis was carried out using a HPLC-DAD (Hewlett-Packard 1100 series equipped with a DAD detector and a Rheodyne injection valve with a 20 µL loop). The chromatographic column was a Spherisorb ODS-2 (250 × 4.6 mm internal diameter and 5 µm particle size) from Teknokroma (Barcelona, Spain) kept at room temperature (25 °C) with a pre-column of the same phase. The eluents were milli-Q water (pH adjusted to 2.5 with trifluoroacetic acid) and acetonitrile (HPLC grade) supplied by Teknokroma. The elution method consisted of the following gradient over a total run time of 55 min: 95% A initially, 75% A in 30 min, 50% A in 45 min, 0% A in 47 min, 75% A in 95 min, 95% A in 52 min until completing the run. The equilibrium time was 5 min and the flow rate was 1.0 mL/min. The chromatograms were recorded at 280 nm. The external standards of 3,4-dihydroxyphenylglycol, tyrosol and hydroxytyrosol were used to identify the compounds by comparing retention times and to quantify them by the regression curves.

HPLC-MS: The phenolic compounds present in the extract obtained from the chromatographic system were identified by electron impact mass data collected on a quadrupole mass analyzer (ZMD4, Micromass, Waters Inc.; Manchester, U.K.). Ionization energies of 50 and 100 eV in negative mode and 50 eV in positive mode were used in the electron spray ionization (ESI) mass spectra. The capillary voltage was 3 kV at 200 ºC as a desolvation temperature; while the source temperature was 100 ºC, and the extractor voltage was 12 V. The flow was 1 mL min−1 in split mode (UV detector MS). The Kinetex^®^ Biphenyl 100 Å C-18 column (250 mm × 4.6 mm i.d. 5 μm) was used. The mobile phases used were 0.01% trichloroacetic acid in water and acetonitrile with the above mentioned gradient for HPLC-DAD.

### 2.8. Determination of Total Phenolics by the Folin-Ciocalteu Method

The procedure described by Obied et al. [20] was used. 80 µL of Na_2_CO_3_ at 0.7 M were added to 20 µL of each of the extracts. Then 100 µL of the 0.2 M Folin solution were added to each well of the plate. After 15 min, readings were taken at 655 nm in a Bio-Rad model iMark microplate reader (Hercules, CA, USA). The results obtained were expressed in gallic acid equivalents, and a calibration line with known gallic acid concentrations was made.

### 2.9. Colorimetric Determination of Total Sugars in the Extracts

The total neutral sugar contents were determined colorimetrically according to the anthrone method [21]. Briefly, each extract was dissolved in an ethanol/water solution (1:1). An aliquot of each dilution of 100 µL (in triplicate) was taken and diluted accordingly and 200 µL of a 0.2% (*w*/*v*) anthrone solution in concentrated sulphuric acid were added. The test tubes were shaken in a Vortex and heated for 5 min at 100 °C in a water bath. When the tubes were cooled in an ice bath the absorbance was measured at 630 nm in the Bio-Rad microplate reader (Hercules, CA, USA). Glucose was used as an external standard to prepare a calibration with aqueous solutions in a concentration range of 0.02–0.2 mg/mL. The results obtained were expressed as mg of glucose per gram of extract.

### 2.10. Determination of Antioxidant Activity In Vitro

#### 2.10.1. Iron Reduction Power

The modification of the method of Psarra et al. [22] was used, using FeCl_3_ as an oxidizing agent. Briefly, the samples were mixed with FeCl_3_ (6 mM in citric acid 5 mMin) in quadruplicate, and incubated for 20 min at 50 ºC. After that, a bipyridyl solution in 1.2% (*w*/*v*) trichloroacetic acid was added, and the plate was read at 490 nm after 30 min using a Bio-Rad microplate reader (Hercules, CA, USA). A calibration curve was established to represent the adsorbance at 490 nm against a known concentration of 6-hidroxy-2,5,7,8-tetramethylchroman-2-carboxylic acid (Trolox) to express the results. The reducing power (PR) was expressed as Trolox equivalents (mg/L) from the equation obtained by linear regression PR = 0.2172 × A490 − 0.180. Measurements of each phenolic extract were made in triplicate together with a blank for each sample.

#### 2.10.2. Anti-radical Activity:2,2-diphenyl-1-picrylhydrazyl (DPPH Assay)

The antioxidant activity of each phenolic extract was determined as the free radical-scavenging capacity using the scavenging of DPPH radical method described in a previous study [23]. The reduction of DPPH by antiradical compounds was measured at 515 nm using a microplate absorbance reader model 550 (Hercules, CA, USA). The radical-scavenging capacity of each extract was expressed as EC50 that represents the concentration in mg/mL at which the oxidation that occurs is reduced by 50%, therefore the lower the EC50 values, the greater the antioxidant capacity of the sample. EC50 was calculated from a calibration curve by linear regression for each extract.

#### 2.10.3. Determination of Antioxidant Activity in Oil: Rancimat Method

The determination of the oxidative stability of sunflower oil enriched with the phenolic extract was carried out using a 679 Rancimat instrument (Metrohm Ltd., Herisau, Switzerland). The enriched and the control oil samples (2.50 ± 0.01 g) were weighed into the reaction vessel directly. The temperature of the heating block was set at 90 °C and the air flow was adjusted to 20 L/h. The results were expressed as induction time (IT), which is the time period in which the secondary oxidation products are detected, expressed in hours. It characterizes the oxidation stability of the oils. All determinations were carried out in duplicate.

The lipid matrix was obtained from sunflower oil by purification through alumina to remove the antioxidant compounds [24]. The antioxidant-free oil was stored at −18 °C under nitrogen gas until use. The enriched oils were obtained by the addition of each extract to the purified sunflower oil, and stirring for 10 min to mix the antioxidants with the oil. The final concentration of the extract was 760 ppm.

### 2.11. Statistical Analysis

The results were expressed as mean values ± standard deviations of replicate experiments. STATGRAPHICS^®^ plus software was used for statistical analyses. Comparisons amongst samples were made using one-way analysis of variance (ANOVA) and the least significant difference (LSD) method at the same confidence level. A significance level alpha or *p* value of < 0.05 indicated statistically significant differences.

## 3. Results and Discussion

### 3.1. Raw Material Analysis

Analysis of the three alperujo water samples (Table 1), corresponding to the three heights (upper, middle and bottom) of the evaporation pond, showed a significant difference in undissolved solid content, increasing with pond depth. The average content was 4.30 g/L, which is not a problem for the use of L/L phenol extraction systems, but is a problem if a chromatographic system is to be used, in which case the use of the corresponding alperujo water would be recommended for the first two heights of the pond where the average solid content does not exceed 2 g/L [19].

The amount of fat increased with the concentration of solids, and the concentration of fat in the liquid was lower on the surface, as in the other two samples. Therefore, the total fat content was higher as the depth of the pond increased. This result is due to the high fat content of the suspended solids, and this solid settles in the pond. The unexpected fact that the fat content in the water is higher at the bottom may be due to the fact that during sample preparation when separating the solids some of the fat contained in the solids passes into the water. It is also important to note that the fat content of the insoluble solid increases with depth, which could be due to oxidative or enzymatic processes favored by an increased presence of light and oxygen causing hydrolysis and oxidation of the lipids in the solid particles. An extraction was carried out by combining the undissolved solids from the three heights and 8.53% of the fat in dry matter was obtained, which is a high enough value to make it feasible to recover this fat at the industrial level through a solvent extraction such as the one usually used in the extraction of alperujo with hexane. It would be recommended to add this solid to the alperujo itself in order to increase its richness in fat and improve the pomace oil extraction process, avoiding problems of solid permeability to the organic solvent as well.

With regard to the total and individual phenols, there was no significant difference among the three samples. The average value of total phenols was 3.92 g/L, which is an average concentration if compared to olive oil mil waste water (OMWW) or three-phase vegetable water, which can reach values of up to 10 g/L [25]. This water came into contact with the alperujo during its storage and came mainly from vegetation water and water from the vertical centrifuge, which had continued to solubilize simple phenols, mainly HT and Ty. The concentration in HT was relatively high, and comparable to the concentration that was obtained after the application of thermal treatment to the alperujo, or much higher than the concentration of HT present in the OMWW [3]. The HT concentration obtained makes alperujo water a very interesting source for obtaining extracts which are rich in this antioxidant.

Other authors have studied the waters of the three-phase extraction system or alpechin and found that the maximum content of phenols is over 1 g/L, that phenols are degraded with the depth of the pond and with the storage time, reaching a loss of 40% after three months. Despite exposure to light and oxygen in the upper layers or possible anaerobic reactions at the bottom, the amount of total phenols and the three main phenols is not significantly altered in the case of alperujo water. This may be due to the high phenolic concentration, four times higher than in the alpechin, which on the one hand inhibits the development of reactions by microorganisms and, on the other hand, protects during the studied storage time against oxidation [26,27].

### 3.2. Phenolic Extracts

#### Characterization of Phenolic Extracts

From the alperujo water, two phenol extractions were carried out by two systems used at the industrial level for the production of HT extracts, by solvent extraction, using ethyl acetate, and by a chromatographic system using ion exchange resin. The values for total and individual phenols as well as total sugars and the amount of extract obtained in each system are shown in Table 2 for the three samples of alperujo water. It can be seen that the amount of extract obtained with solvent was much higher than with resin; while the concentration of phenols and sugars was lower, which showed that chromatography was a more selective process. With respect to the depth of the pond, significant differences can be seen in the amount of extract obtained at the surface and at the bottom for the ethyl acetate extraction as well as significant differences in the concentration of phenols in the extract of top A where both the total phenol content and the concentration of DHPG were significantly higher than in the other extracts. However, there were no differences in the concentration of HT or Ty.

In the case of the extracts obtained chromatographically there were no significant differences in the amount of extract obtained or the content in total sugars for the three samples studied. However, there was a lower concentration of HT in the extract of the upper part A, in contrast with the content in total phenols, which was higher. In the same way, the extract which had a higher quantity of DHPG, twice the amount of the rest, was the one obtained from the alperujo waters in the lower part C. Therefore, the extraction of HT and DHPG was higher at the bottom, which may be due to the fact that the pH in that zone is 5.1, slightly higher than that of the upper and middle parts of 4.5, which favors the adsorption of these simple phenols in the ion resin [19].

Figure 1 shows a mass balance comparing the average extraction from the three depths with the two systems. The solvent extraction produced 3.4 times more extract, which had a lower phenol concentration. The amount of HT was two times higher in the case of the ionic resin extraction. Therefore, the extraction with resin was more selective, although to improve this system it would be necessary to reuse this source several times to extract more phenols. Reuse, together with the environmental advantages of using resins, currently makes this system the most widely used in the industry. From one ton of alperujo water it is possible to obtain 12.7 and 3.7 kg of dry extract with 5.12 and 7.84% of HT and 0.2 and 0.08% of DHPG for solvent and resin extraction systems, respectively. In both cases, this means 5.4 kg of solids containing 1.1 kg of oil and 4.3 kg of solids.

Figure 2 shows the chromatographic profiles of the alperujo water and the two extracts. The profile of the alperujo water was identical to that of the extract obtained with ethyl acetate; only the first peaks at short retention times, corresponding to the organic acid zone, disappeared. However, in the profile of the extract obtained with ionic resin, other peaks appeared after tyrosol in higher concentrations. As the extract obtained with resin was more concentrated in phenols, it was used to identify the rest of the phenols.

Therefore, this extract was subjected to fractionation and analysis by liquid chromatography with a mass detector. The extract was dissolved in water and fractioned in the same ion exchange resin column, obtaining ten fractions from F1 to F10. The most representative fractions were analyzed by HPLC-MS to identify their main phenolic compounds. Table 3 shows the results of the identifications and the presence of a large number of phenol derivatives which are abundant in alperujo, coming from the chemical and enzymatic processes which occurred during the storage of alperujo. This profile is similar to the ones found by other authors in olive oil waste water, although in lower concentrations [28,29]. The components with higher molecular weights appeared in the same part of the spectrum and with the same form as polymeric phenolic fractions appeared, according to other authors, which could mean that part of these polymerized phenols was solubilized in the alperujo water [30,31].

### 3.3. Antioxidant Activity

The anti-radical potential, reducing power and antioxidant capacity in oil were determined in vitro to determine the potential application of the extracts obtained (Figure 3 and Table 4).

#### 3.3.1. Anti-Radical Potential (DPPH)

The results are expressed as EC_50_ (Figure 3), which is the concentration of the extracts in mg/mL necessary to inhibit oxidation by 50%, so that the more anti-radical capacity it has, the smaller the amount of it is required to reduce this oxidation by 50%. There was a significant difference between the two groups of extracts obtained by the two systems, with the free anti-radical activity in the phenolic extracts extracted with resin being higher. This may be due to the fact that there was a higher amount of simple phenols, as in the case of hydroxytyrosol, and of total phenols in the extracts obtained with the resin, compared to the extraction with ethyl acetate, in which the extract obtained from the upper sample had significantly higher activity, probably because of its higher concentration in DHPG.

The EC_50_ values were below those found in the literature [8], which were obtained with a hydroxytyrosol extract of 50% purity in dry weight, corresponding to an EC_50_ of 3.8 mg/mL. The concentration of HT and DHPG contained in the alperujo water extracts was between 0.16 and 0.008 mg/L, which was in the same order as the EC_50_ of pure HT according to other authors of 0.11 mg/mL, and well below that found for pure DHPG of 0.17 mg/mL [32]. Therefore, it would be due to the presence or absence of other compounds besides hydroxytyrosol which presented activity and could increase or decrease the anti-radical character of the obtained extracts.

#### 3.3.2. Reducing Power (RP)

The results for reducing power show that the activity increased with the depth at which the samples were taken, and that the values were higher in the case of extracts obtained with resin, except for the central zone of the pond. In this case, it was not possible to establish a correspondence with the concentration of simple phenols, with an inverse relationship with the content in total phenols, which decreased slightly. Therefore, the presence of other types of compounds must have justified the activity shown in these tests, whether they were of phenolic origin or other metabolites formed fundamentally at the bottom of the pond during storage.

In this case, the values obtained with these extracts were lower than those shown by the pure HT and DHPG in other works [32]. This is due to the fact that these phenols were found in lower concentrations in the alperujo water extracts. The data obtained would then be compared to those obtained with a natural and a commercial extract of high purity in DHPG, with lower values than the ones shown by other authors for hydroxytyrosol extracts which exceed 50% purity in weight [8].

The higher reducing power observed in the samples from the bottom of the pond may be due to the presence of metabolites formed in a lower concentration of oxygen, or by oxidative degradation at the surface or in the middle zone of the pond. In the case of oxidative degradation, long periods of storage should be avoided, as this may reduce the activity of the extracts and at the same time produce unpleasant odors.

#### 3.3.3. Lipid Oxidation (Rancimat Test)

Protection against rancidity, as a measure of antioxidant activity in a lipid matrix (sunflower oil), was tested by the Rancimat method. The three extracts obtained with ethyl acetate on the one hand, and the three extracts obtained with resin on the other hand, were mixed, dried and dissolved completely in sunflower oil which had been previously depleted of its antioxidants.

It can be seen in Table 4 that the addition of the ethyl acetate and resin extracts to the oil increased the oxidation induction time by more than 23 and 34%, respectively. Again, the chromatographic extract showed increased activity. Both extracts showed good protection against lipid oxidation of the oil despite low concentrations of HT and DHPG. These results are similar to the values obtained in another work, where a 33% increment in the induction time was obtained using a natural extract which contained 50 ppm of HT in the same edible oil [13]. The same relationship between the increase in oxidation time and the concentration of HT was seen in both extracts. This seems to indicate that the concentration of HT was one of the most important factors to justify this activity. It should also be taken into account that the presence of DHPG improved the HT activity as had already been determined in the above mentioned work where the synergism of the two phenols was established. Therefore, the extracts obtained could serve to improve the shelf-life of refined oils both for direct consumption and for frying, or for heterogeneous foods based on oil emulsions.

## 4. Conclusions

A liquid phase from the water contained in stored alperujo has appeared in industry, because until now it was homogenized with the alperujo for its treatment. This liquid phase contains a substantial amount of bioactive components such as phenols, among which, HT and DHPG stand out for their biological importance and concentration. The storage period can lead to a variation in the concentration of these phenols and suspended solids depending on the depth of the pond. The presence of solids with a high fat content would allow the recovery of this fat or increase the concentration of fat in the alperujo for its extraction. The application of two industrial processes in the present work shows that it is possible to obtain phenolic extracts with high phenol concentration and antioxidant activity at the industrial level. The extraction of phenolics is gaining importance in the food industry; hence the cost of these extracts is reduced [18]. The extracts can be used for oxidation protection in both aqueous and oily matrices, helping not only to revalue this effluent but also to improve the formulation of new foods based on natural antioxidants, or for nutraceutical or pharmaceutical purposes [13]. It is also important to note that the elimination of phenols from alperujo water would improve the purification of this effluent by promoting the use of this system as a biorefinery, which minimizes its environmental impact [18].

## Figures and Tables

**Figure 1 foods-09-00962-f001:**
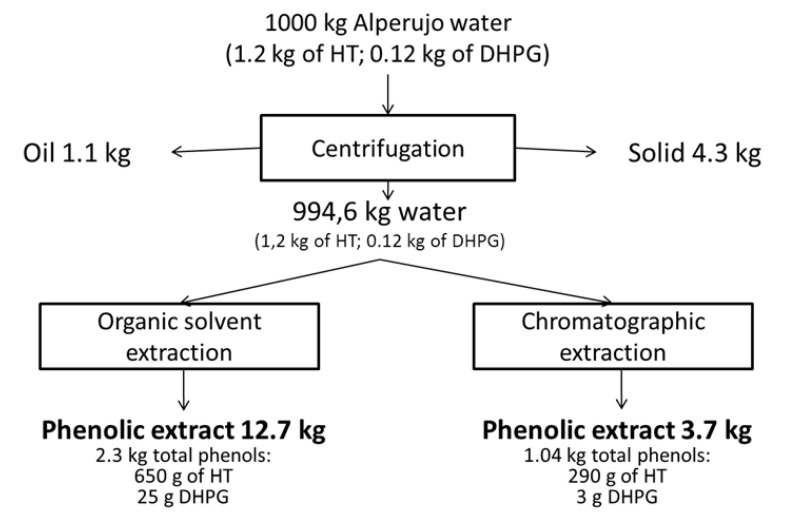
Balance of the two extraction systems from alperujo water referring to the overall extract from the three depths.

**Figure 2 foods-09-00962-f002:**
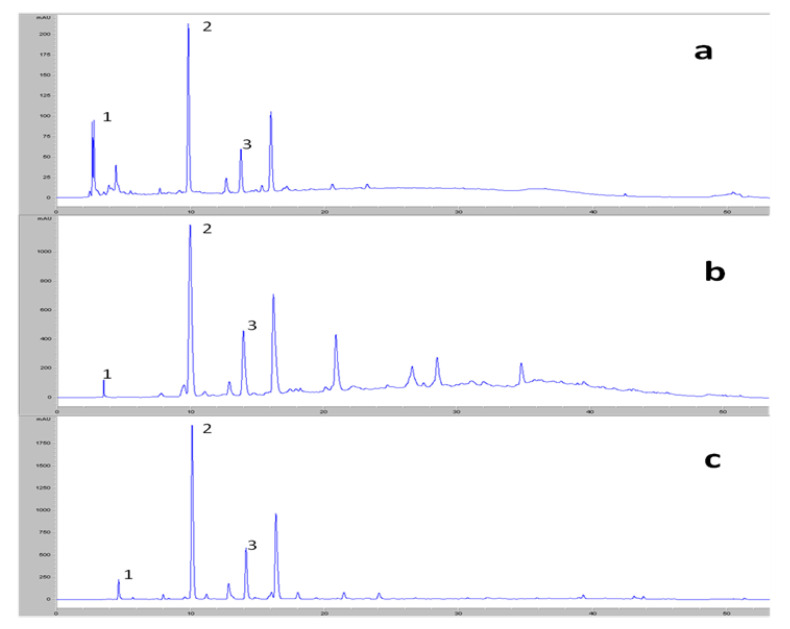
Profiles obtained by HPLC at 280 nm for a sample of alperujo water (**a**), the phenolic extract obtained with resin (**b**) and the phenolic extract obtained with ethyl acetate (**c**), identifying DHPG (1), hydroxytyrosol (2) and tyrosol (3).

**Figure 3 foods-09-00962-f003:**
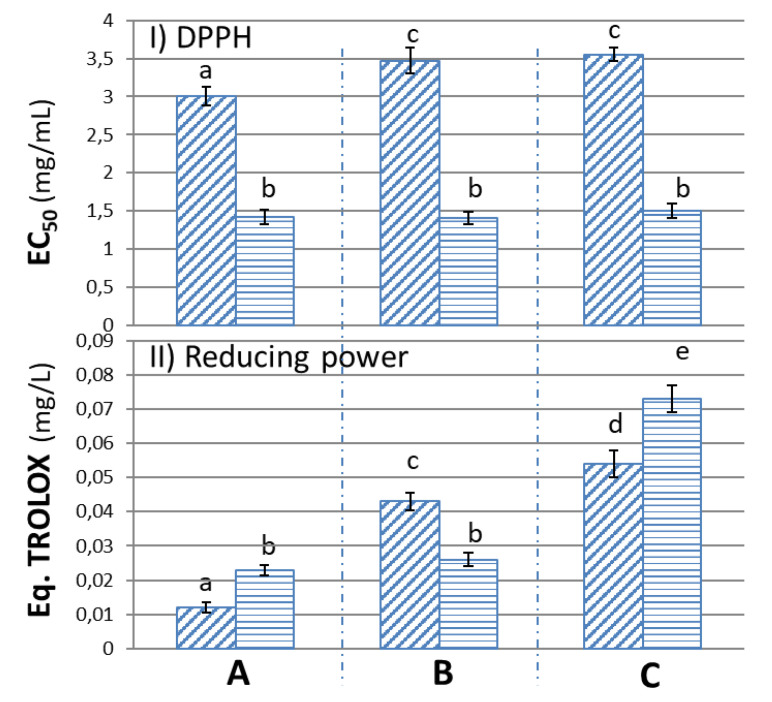
Activities determined by DPPH (**I**) and reducing power (**II**) methods of each phenolic extract obtained by solvent (oblique lines) or resin (horizontal lines). Methods I and II were carried out using the three samples obtained from different depths of the pond (upper **A**, middle **B** and lower **C**). Means with the same letters were not significantly different, *p* < 0.05.

**Table 1 foods-09-00962-t001:** Chemical parameters, suspended solid, fat and phenolics of the alperujo water collected from three different heights of the pond, upper (A), medium (B) and down (C).

Chemical Parameters		Different Heights of the Pond		
		A (Upper)	B (Medium)	C (Down)
Suspended solid (g/L)		1.71 ± 0.06c *	2.23 ± 0.10b	8.49 ± 0.38a
Fat concentration (g/L)				
Fat in solid		0.15 ± 0.03c	0.19 ± 0.02b	0.77 ± 0.13a
Fat in water		0.59 ± 0.11b	0.77 ± 0.07a	0.70 ± 0.06b
Total fat		0.74 ± 0.14c	0.96 ± 0.09b	1.47 ± 0.19a
Phenolic compounds (g/L)				
Total phenols		3.74 ± 0.41a	3.91 ± 0.17a	4.11 ± 0.22a
Individual phenolic	3,4-dihydroxyphenylglycol	0.11 ± 0.02a	0.12 ± 0.01a	0.12 ± 0.01a
	Hydroxytyrosol	1.12 ± 0.09a	1.27 ± 0.04a	1.21 ± 0.01a
	Tyrosol	0.27 ± 0.04a	0.33 ± 0.03a	0.31 ± 0.03a

* Means with the same letter in the same row were not significantly different, *p* < 0.05.

**Table 2 foods-09-00962-t002:** Chemical parameters, grams of extracts, sugar, total and individual phenolics of the alperujo water extracts collected from three different depths of the pond, for the two extracts obtained from industrial process. Main individual phenolic compounds: 3,4-dihydroxyphenylglycol (DHPG), hydroxytyrosol (HT) and tyrosol (Ty).

Chemical Parameters of the Extracts		Different Depths of the Pond		
		A (Upper)	B (Medium)	C (Down)
Ethyl acetate extract				
g of extract/L of alperujo water		10.65 ± 1.43b *	12.95 ± 2.02ab	14.44 ± 1.77a
Sugars	Total sugar (g/g of dry extract)	0.23 ± 0.01a	0.19 ± 0.01a	0,19 ± 0,02a
	% referred to dry matter	2.29	1.93	1,94
Phenolic	Total phenols	0.19 ± 0.07a	0.17 ± 0.03b	0,17 ± 0,04b
	(g/g of dry extract)			
	% referred to dry matter	19.42	17.21	17,12
Individual phenolic(mg/g of dry extract)	DHPG	2.71 ± 0.30a	2.04 ± 0.05b	1,21 ± 0,12c
	HT	52.91 ± 3.14a	48.37 ± 4.16a	53,23 ± 1,77a
	Ty	24.04 ± 2.33a	18.19 ± 2.07a	20,23 ± 3,91a
Chromatographic extract				
g of extract/L of alperujo water		3.39 ± 0.12	4.05 ± 0.06	3.65 ± 0.19
Sugars	Total sugar (g/g of dry extract)	0.33 ± 0.02a	0.28 ± 0.03a	0.34 ± 0.04a
	% referred to dry matter	3.31	2.83	3.35
Phenolic	Total phenols (g/g of dry extract)	0.33 ± 0.04a	0.26 ± 0.02b	0.26 ± 0.03b
	% referred to dry matter	33.44	26.18	26.25
Individual phenolic (mg/g of dry extract)	DHPG	0.64 ± 0.02a	0.53 ± 0.06b	1.10 ± 0.11b
	HT	64.89 ± 3.01a	89.27 ± 5.45b	82.84 ± 4.67b
	Ty	34.38 ± 2.18a	31.11 ± 2.50a	32.84 ± 3.07a

* Means with the same letter in the same row were not significantly different, *p* < 0.05.

**Table 3 foods-09-00962-t003:** Phenolic compounds identified by HPLC-DAD-MS in the fraction obtained from the phenolic extract by the ionic resin column in comparison to the results obtained by Rubio-Senent et al. [22].

Retention Time (min)	Compounds	Molecular Weight	λmax (nm)	m/z ^−^
10.20	Hydroxytyrosol	154	214, 234, 278	153, 123
14.25	Tyrosol	138	200, 218, 275	137
15.41	Elenolic acid derivative	242	230	241, 237, 135
16.80	Tyrosol derivative	186	275	185, 151, 137
18.10	Vanillic acid	168	200, 218, 255, 298	167, 108, 45
21.23	Acido 4-Hidroxibenzoico	336	278	335, 215, 153, 125
23.60	Hydroxytyrosol derivative	234	278	233, 151, 123
24.78	3,4-Dihydroxyphenylacetic acid derivative	168	214, 234, 278	151, 123, 109, 59
25.53	4 metilcatecol	124	236	123, 107, 69
27.05	Hydroxytyrosol derivative	211	280	210,151, 123, 59,
29.35	Luteolin-7-O-Rutinoside	594	200, 254, 349	593, 447, 285, 151
30.13	Verbascoside	624	198, 328	623, 461, 161
31.92	Oleuropein aglycone derivative	378	200, 222, 280	377, 225, 123
32.58	Oleuropein derivative 1	538	214, 234, 278	537, 377, 287, 257, 211, 123
34.09	Oleuropein derivative 2	378	280	377, 361, 313, 180, 151
36.02	Oleuropein derivative 3	538	280	537, 403, 385, 223, 151, 123
38.07	Unidentified	630	280	629, 303, 187, 123
39.68	Unidentified	630	280	629, 303, 185, 183, 139
39.97	Unidentified	630	280	629, 305, 186,185
42.51	Unidentified	810	280	809, 613, 563, 359, 195, 123,153
42.97	Unidentified	810	280	809, 323, 195, 125,
45.00	Unidentified	810	280	329, 307, 185, 125
46.23	Unidentified	810	280	329, 285, 187, 153

**Table 4 foods-09-00962-t004:** Activity determined by Rancimat method of each phenolic extract obtained by solvent or resin from the mixture of the three samples. The induction time of each extract was statistically compared.

Rancimat Test		Concentration (mg/L)	Induction Time (h)	ΔIT (%)
Control (without extracts)		0	7.30 ± 0.03 b *	-
Solvent extract		760	9.23 ± 0.18 a	23.4
Main phenols	DHPG	1.5		
	HT	39.1		
	Ty	15.8		
Resin extract		760	9.85 ± 0.17 a	34.9
Main phenols	DHPG	0.6		
	HT	60.0		
	Ty	24.9		

* Means with the same letter in the same column were not significantly different, *p* < 0.05.

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
