# Peer review of "New Liquid Source of Antioxidant Phenolic Compounds in the Olive Oil Industry: Alperujo Water"

_foods, 2020, doi:10.3390/foods9070962_

Round 1

Reviewer 1 Report

The Authors significantly changed the content of the article, which definitely improved it. The significance and purpose of the research were described.
The exact manner in which the research material was collected was given. However, it is still necessary to specify how and where the samples were kept. This applies to information in lines 121-122 "... months of the campaign, so the average storage time was two months. All samples were kept at -20 ºC until use."

In lines 299-301: "... Therefore the extraction of HT and DHPG was higher at the bottom, which may be due to the fact 300 that the pH in that zone is 5.1, slightly higher than that of the upper and middle parts of 4.5, which favors the adsorption of these simple phenols in the ion resin. " But it was not given how the pH was determined and was it done at all? Because this information is lacking, it is difficult to say unequivocally that the pH of the solution in the upper layer was 5.1, as there is no effect given in the method. The authors are asked to complete this information in chapter 2. Materials and methods. In addition, this conclusion should be referred to in the literature.

In chapter 2. Materials and methods, there is no information on the method for determining the dry matter content of the samples. And in the results listed in tab. 2 sum of sugars and phenols is given in dry matter. The reference to% in terms of dry matter is not very clear and incomprehensible. This should be harmonized and the methodology for dry matter given, especially since the authors indicate in lines 131-134 when collecting: Approximately one liter of each sample was spun in six cups. The exact amount was not indicated, the wording around does not allow to accurately determine the amount of solid obtained separated from the liquid.

Please indicate whether a normal analysis was carried out to verify the interpretation of the results obtained before carrying out the statistical analysis. If so, please indicate how and using which actions and provide the necessary data for this analysis. If no, please explain how the compatibility and correctness of interpretation of the results obtained are provided. In addition, it is worth including the level of significance at which the data were analyzed.

The captions of the tables must be correctly changed according to the magazine's recommendations. In addition, it is necessary to replace commas (,) with periods (.) In all tables.
The tabular data in Fig. 3 should be edited to the normal table, because it is. invalid signature. I understand that the authors wanted to authenticate their data, but they should be given in the table under Fig. 3. Similar suggestions will be shown in Fig. 4, where the table should be a separate representation because it is not uniform with the figure. Also explain all abbreviations used in the tables and figures.

Author Response

Thanks for the reviewer´s comments.

The Authors significantly changed the content of the article, which definitely improved it. The significance and purpose of the research were described.

The exact manner in which the research material was collected was given. However, it is still necessary to specify how and where the samples were kept. This applies to information in lines 121-122 "... months of the campaign, so the average storage time was two months. All samples were kept at -20 ºC until use."

Response: The following phrase has been insert in the text to explain how and where the samples were kept (line 120-121): “The samples were all transported on the same day they were taken and stored in 1 and 5 L containers in freezers at - 20 ºC until use”.

In lines 299-301: "... Therefore the extraction of HT and DHPG was higher at the bottom, which may be due to the fact 300 that the pH in that zone is 5.1, slightly higher than that of the upper and middle parts of 4.5, which favors the adsorption of these simple phenols in the ion resin. " But it was not given how the pH was determined and was it done at all? Because this information is lacking, it is difficult to say unequivocally that the pH of the solution in the upper layer was 5.1, as there is no effect given in the method. The authors are asked to complete this information in chapter 2. Materials and methods. In addition, this conclusion should be referred to in the literature.

Response: A new section (2.5) in material and method has been inserted to explain how the pH was measured (line: 146-150). The influence of the pH into the resin absorption has been referred to the literature: “[19]”.

In chapter 2. Materials and methods, there is no information on the method for determining the dry matter content of the samples. And in the results listed in tab. 2 sum of sugars and phenols is given in dry matter. The reference to% in terms of dry matter is not very clear and incomprehensible. This should be harmonized and the methodology for dry matter given, especially since the authors indicate in lines 131-134 when collecting: Approximately one liter of each sample was spun in six cups. The exact amount was not indicated, the wording around does not allow to accurately determine the amount of solid obtained separated from the liquid.

Response: It is also true, thus, the description of the determination of dry matter has been inserted in the new section (2.5) in material and method (line: 146-150): “2.5. Determination of pH and the amount of dry matter content of the samples”.

With regards to the determination of suspended solid the following words has been insert to clarify how the quantification was made (line 130): “measuring exactly each volume”.

Please indicate whether a normal analysis was carried out to verify the interpretation of the results obtained before carrying out the statistical analysis. If so, please indicate how and using which actions and provide the necessary data for this analysis. If no, please explain how the compatibility and correctness of interpretation of the results obtained are provided. In addition, it is worth including the level of significance at which the data were analyzed.

Response: according with this comment the statistical section has been modified (line: 237-240): “The results were expressed as mean values ± standard deviations. STATGRAPHICS® plus software was used for statistical analyses. Comparisons amongst samples were made using one-way analysis of variance (ANOVA) with Student’s t test and the LSD method at the same confidence level. A p value of < 0,05 indicated statistically significant differences”.

The captions of the tables must be correctly changed according to the magazine's recommendations. In addition, it is necessary to replace commas (,) with periods (.) In all tables.

Response: It has been improved in the tables and the commas also replaced with periods.

The tabular data in Fig. 3 should be edited to the normal table, because it is. invalid signature. I understand that the authors wanted to authenticate their data, but they should be given in the table under Fig. 3. Similar suggestions will be shown in Fig. 4, where the table should be a separate representation because it is not uniform with the figure. Also explain all abbreviations used in the tables and figures.

Response: Figure 3 has been changed by table 3 because all information is on the table. The Figure 4 has been separate in Figure 4 and Table 4 in agreement with the reviewer comments.

Reviewer 2 Report

No other comments to add. The authors improved in significant way the manuscript.

Author Response

Reviewer 2.

Thanks for the reviewer´s comments.

Response to the reviewer ´s comments.

All the corrections made in the manuscript have been showed in red.

Reviewer 3 Report

Responses to previous comments were satisfactory in all cases.

There's a minor issue on Lines 71-73, the sentence it's not clear. Also, I guess it is not an h between the brackets but a reference number. 

In Tables 1 and 2, if there is not a statistically significant difference, it is not necessary to include the same letter in the entire row. 

Author Response

Reviewer 3.

Responses to previous comments were satisfactory in all cases.

Thanks for the reviewer´s comments.

There's a minor issue on Lines 71-73, the sentence it's not clear. Also, I guess it is not an h between the brackets but a reference number.

Response: the sentence has been improving by clarifying the storage in the open air ponds (line 72). The “h” has been replace by it reference number.

In Tables 1 and 2, if there is not a statistically significant difference, it is not necessary to include the same letter in the entire row.

Response: The letters in the rows where there was no significant difference in both Table 1 and Table 2 have been removed.

Reviewer 4 Report

I carefully read the response of authors to the issues raised by referees.

Above all, I believe that there is a difference between a scientific paper and a technical report, and I believe that authors wrote a good techical report for a specific customer, though representing the largest "producer" of alperujo.

This research has serious flaws in the experimental design, lacking of representativeness of data variability. Sampling is subject to a number of uncontrolled variables (initial characteristics of the alperujo as a function of olive quality, sampling time and sedimentation, and even evaporation with a consequent concentration of solutes). None of these variables was considered in the experimental design. Therefore, the significance of the results for readers other than the owner of the (three) pond(s) is dramatically reduced.

Neither information about the equipment adopted for counter current extraction is sufficiently provided.

Therefore, the data provided are useful "hic et nunc", here and now, and this is, in my opinion, at the opposite of scientific dissemination.

Author Response

Reviewer 4.

I carefully read the response of authors to the issues raised by referees.

Thanks for the reviewer´s comments.

Above all, I believe that there is a difference between a scientific paper and a technical report, and I believe that authors wrote a good techical report for a specific customer, though representing the largest "producer" of alperujo.

Response: the authors do not agree with this comment. In my case, I have been working on the use of effluents and by-products of the olive oil industry for more than 20 years. This research work has been carried out not for one company but to offer the whole sector a solution through the extraction of high added value components and to avoid the environmental problems that are starting to appear with this effluent. In fact, thanks to this work we can eliminate many evaporation ponds and obtain a valuable source of hydroxytyrosol and DHPG much more easily manageable than those that were being used until now.

This research has serious flaws in the experimental design, lacking of representativeness of data variability. Sampling is subject to a number of uncontrolled variables (initial characteristics of the alperujo as a function of olive quality, sampling time and sedimentation, and even evaporation with a consequent concentration of solutes). None of these variables was considered in the experimental design. Therefore, the significance of the results for readers other than the owner of the (three) pond(s) is dramatically reduced.

Response: As has already been argued and written in the manuscript, sampling has been carried out in the most representative ponds, covering more than 25% of the world's production. It is very difficult to find works where the representativeness is so high for a sector like the food sector. The sampling has been carried out in the three largest factories in the world, with an adequate sampling in the middle of the season, ensuring the representativeness of the samples and of the evaporation. Precisely the coordination, the sampling and the choice of the ponds has been designed so that the results are totally representative and comparable to the rest of the industries.

Neither information about the equipment adopted for counter current extraction is sufficiently provided.

Response: The adequate information about this equipment was added after the first revision, and added to material and method section: (Alamo-Ref.:00362000, Spain). In addition, the following sentence has been insert in the text (line: 157-158):” Basically, the solvent, upon evaporation, passes from the bottom of the sample to the top, back to the heat source where it is again evaporated”.

Therefore, the data provided are useful "hic et nunc", here and now, and this is, in my opinion, at the opposite of scientific dissemination.

Thank for the reviewer point of view, but research work of this kind is what has helped us to change the olive oil sector, and this work will undoubtedly help us to improve the sector even more while allowing us to find a source of DHPG, which is more difficult to find than hydroxytyrosol, which is very important because of the properties of this new phenol and the synergy it has with the rest of the phenols.

Round 2

Reviewer 4 Report

I expressed my concerns regarding this manuscript. I received authors' responses. I respect their point of view, though I do not agree with them.

I have no doubt that the work of the authors in the last decades changed the olive oil sector. Working in technological transfer is really worth of the highest consideration, in my opinion. I only believe that not always the results of a huge work find their best outcome in a scientific paper.

In my opinion, the results of research can find different outcomes, and the publication of a research paper is not necessarily the only one nor the best.

I believe that publishing a research paper requires that the work should be replicable by other authors and in other contexts.

I believe that this is a very good work from a technical point of view, but I do not see how it can be considered replicable by other authors and in other contexts.

Therefore, my concern is on epistemolgical basis and not on technical basis. I am sure that Editor will take the right decision, considering, together with my considerations, the suggestions by other referees.

Author Response

I respect and appreciate the reviewer's comments. Our long experience in this sector tells us that this type of work helps not only the olive oil sector, but also other agro-industrial by-products where the techniques we use are replicated in other works by other authors, which is helping to improve the recovery of waste such as cocoa, dates, strawberry or tomato among others following the achievements we have made and published in this way.

This manuscript is a resubmission of an earlier submission. The following is a list of the peer review reports and author responses from that submission.

Round 1

Reviewer 1 Report

This paper deals with the application of industrial extraction processes to recovery phenolic compounds from alperujo water. Though the subject is interesting, the paper has some major drawbacks.

The extraction procedures are not fully described and are partially covered by patents. Therefore, the comparison of extraction techniques is of limited interest for readers, in my opinion. The aspect of novelty I see regards the focus on alperujo water from ponds. I also appreciated to approach at industrial level.

Nevertheless, few information is provided about the representativeness of the sampled waste and no source of potential variability (except sampling depth) is considered or covered: what was the volume of processed olives from which the alperujio water derived? How long alperujo water rested before sampling? Is there any effect of the phase of harvesting season in which alperujo water is submitted to extraction? I believe that when evaluating a new potential source of compounds of interest, sampling should cover at least some potential sources of variability. On the other hand, only one pond was sampled. As a consequence, I am concerned about the representativeness of this study, that should be considered of preliminary nature. This should appear in the title and underlined in the text.

Please see also the following issues:

Abstract

  • The first sentence is not clear. Please rephrase.

Introduction

  • This section contains only self-citations of the authors. The Introduction should be rewritten considering the contributions of authors other than the authors of the present paper.
  • In general, I found 14 self-citations in 22 total references of this manuscript (63.6%). I strongly suggest to critically limit self-citation evaluating appropriate cases, and contextualize the research considering the whole scientific panorama, besides authors’ own papers.

Materials and methods

  • Lines 75-78. I believe that some relevant information is missing. Authors sampled at different heights, hypothesizing that sedimentation phenomena would affect the concentrations of bioactive compounds as a function of depth of sampling. In sedimentation phenomena, time is a relevant variable. Therefore, authors should provide information about time of sedimentation. How long alperujo was collected in the pond, and how long was it left to rest before sampling?
  • Moreover, how deep was the pond? At what heights sampling was carried out?
  • Line 87. Replace “litter” with “liter”
  • Lines 105-108. Could authors provide details about the counter-current extraction system (e.g. model, producer…)?
  • Line 142. Replace “contractions” with “concentrations”
  • Line 144. Replace “antrone” with “anthrone”

Results and discussion

  • Table 1. Please check lettering for LSD comparisons (see, e.g., the lettering of fat in water)
  • Lines 199-207. The trends in fats in solids and fats in water is unexpected, in my opinion. I invite authors to discuss these data and not only data of total fats
  • Table 2. Please check lettering for LSD comparisons (see, e.g., the lettering of g/L of chromatographic extract)
  • Figure 1. Does the figure refer to the overall extract from the three depths?
  • Figure 4. Please add standard deviations to bar charts
  • Lines 259 and 313. Replace “DHFG” with “DHPG”
  • Lines 318-319. Are authors sure that unpleasant odours are due to oxidative degradation rather than to anaerobic reductive phenomena?
  • Lines 330-334. Since the extracts are mixtures of different phenolic compounds, in some cases unidentified, it is not clear how authors relate the antioxidant activity specifically to HT rather than other phenolic compounds, even with o-diphenolic structures.
  • Lines 348-350. This sentence is speculative. I believe that authors could advance no more than a hypothesis requiring verification.
  • On the whole, while the comparison of extraction techniques has been discussed, the comparison of different depths has been carried out with a descriptive approach, with a poor discussion of the observed differences. Discussion of this aspect should be enriched, at least with some hypotheses, as authors did in lines 315-317.

Reviewer 2 Report

I have read the article "New liquid source of antioxidant phenolic compounds in the olive oil industry: alperujo water". The article describes the potential use of a liquid effluent of the olive oil industry as a source of antioxidant compounds for the food industry, in particular hydroxytyrosol, 3,4-dihydroxyphenylglycol, and tyrosol. After a complete analysis, I have encountered two main issues with this manuscript.

First, the scientific question that the researchers try to respond to is not clearly stated, as well as the aims of the study. While the data presented might be of interest, it lacks appropriate structure, clarity, and an overall connective idea. 

Second, the industrial purification of the phenolic compounds from the Alperujo water, which is of certain interest, is not adequately described in a way that can be reproduced by other researchers. 

Title

The olive oil industry has been using two-phase decanters and storing olive pomace in pools for decades now. Alperujo water is not a new source of phenolic compounds.

Abstract

Lines 12-14: This sentence is incomprehensible.

Line 16: Units

Introduction

Line 29-30: It is relevant to introduce the not specialized reader to the olive oil extraction process and how the two and three phases systems differ.

Line 31: Alperujo is a Spanish word; it is advisable to include a description of what Alperujo is before in the text, even in the abstract of the article.

Lines 69-72: The specific question the research is trying to respond as well as clearly defined aims should be stated. These statements are too broad and unspecific. Which would be the impact of the research?

Materials and methods

Lines 75-78: How was the sampling performed? Were replicated samples taken? Which volume was each sample? A precise description of the sampling procedure is required in order for the experiment to be replicated by other researchers.

Line 84: Format of chemical formulas.

Lines 103-112: The description of the extraction systems is too weak; there's no scale indicated, not the amount of sample treated, nor how the samples were collected for the industrial purification.

Line 185: Were correlation coefficients used for the data analysis?

Results and discussion

Table 1: The table is not well organized. Formatting is poor.  

Table 2: Same as Table 1

Figures 1, 2, 3, and 4: Tables should be independent of Figures. The format should be consistent among Tables and Figures. Labels in Figure 1 are not legible.

Conclusions

Line 338: The two-phase decanter has been used by the olive oil industry for years. The Alperujo water has not appeared now. This statement is not valid. 

Reviewer 3 Report

The recovery of functional compounds from industry waste is becoming an argument of increasing interest. the recovery of polyphenols from alperujo water can represent a valid proposal to be applied from olive industry , also to avoid, as more as possible, the pollution and to save the environment. I think that the alperujo water can be source not only of polyphenols, of course. For this reason, I suggest to the authors to deeply analyse also, in next future, the fat profile of the alperujo water, so to investigate if the fat portion of this product can be source of fatty acids with healthy properties to be used in the formulation of functional ingredients.

Did you calculate the cost for the extraction of these biomolecules? I think such aspect should be considered: if you want propone this methodology for the industrial exploitation, the industry should know also what can be the rate between cost and benefits.

Reviewer 4 Report

Research is interesting but incomprehensible. It is difficult to state clearly which leachate was tested and for what purpose. A number of results included in tables cut out from computer systems do not ensure readability and transparency. There are no methods that could be considered as commonly used in the determination of active compounds in aqueous solutions. The authors do not explain properly why they dealt with this topic. On the one hand, we get information about caring for the environment, on the other, a brief explanation of the problem.
The conclusions do not suggest what conclusions result from the experiment. These are only statements of results obtained during the experiment. There is no data in the method of chromatographic analysis on how the content of identified compounds was counted. The sum of phenols was determined, but the content of the identified substances was not reported using the HPLC DAD and HPLC MS methods. At this stage, it would be worth providing the content of identified compounds. Similarly, with the fat content, which was determined only quantitatively, the most important issue is what fatty acids are in the leachate. A profile alone is not enough. Authors should consider supplementing these missing elements. Without them, work is another study confirming that leachate contains certain amounts of active substances. But it cannot be inferred from this that the extracts obtained can be used to improve the shelf life of refined oils both directly for consumption and for frying or for heterogeneous food based on oil emulsions. Undoubtedly, the work has practical potential, but the authors have skilfully presented these possibilities. It is necessary to supplement with specific issues that speak for the effectiveness and expediency of the experiment. In its current form it is difficult to see.

It is even a dismissive treatment of the topic and the criteria for the preparation of the article, this applies to both tables, charts and interpretation of the presented results. The article as it stands. does not bring significant value to the existing state of knowledge. It is only a confirmation of known facts.